# We Don’t Talk about It: Cancer Pain and American Indian Survivors

**DOI:** 10.3390/cancers12071932

**Published:** 2020-07-17

**Authors:** Felicia Schanche Hodge, Tracy Line Itty, Christine Samuel-Nakamura, Mary Cadogan

**Affiliations:** 1School of Nursing, University of California, 700 Tiverton Avenue, Room 5-934A Factor Building, Los Angeles, CA 90095-1702, USA; tracy.itty@gmail.com (T.L.I.); csamnak@ucla.edu (C.S.-N.); mcadogan@sonnet.ucla.edu (M.C.); 2Fielding School of Public Health, University of California, 640 Charles E Young Dr. South, Los Angeles, CA 90024, USA

**Keywords:** American Indian, cancer pain, cancer survivors, cultural competency, communication

## Abstract

Pain is a common symptom among cancer survivors, yet is rarely talked about by American Indians. Understanding the reasons for reduced communication by American Indian cancer survivors is important for healthcare providers, family members, and others providing treatment and support for cancer symptoms. Thirteen focus groups with Southwest American Indian adult cancer survivors were audiotaped and transcribed as part of a randomized intervention to remove barriers to cancer symptom management. Constant comparative methods were employed in the data analysis, topic categories were grouped for comparison, and final assessment followed Grounded Theory methods. Findings were categorized into two major groupings: communication with family members and communication with health care providers. Within these two groupings, three themes emerged to describe cancer pain experiences and communication barriers: (1) We don’t talk about it, (2) Respect for healthcare providers; and (3) Culturally prohibitive topics on death and pain experiences. Not talking about their cancer diagnosis and cancer-related pain leaves many American Indian cancer survivors without much-needed social support, contributing to reduced treatment compliance and access to healthcare. Findings have implications for educational interventions and quality of life improvement for American Indian and other underrepresented communities.

## 1. Introduction

Understanding the communication patterns of American Indian cancer survivors is important for the mitigation and control of cancer-related symptoms. With knowledge of the communication styles of American Indian cancer survivors, healthcare providers may better mitigate and treat cancer-related pain. Cancer pain can be chronic and the association between depressive disorders and pain has contributed to the increased need for medical attention [1]. Although cancer pain can be managed to a degree, it is not always managed effectively or on a timely basis [2]. Pain management is traditionally under the care of the medical provider, who can provide a medley of pharmaceuticals. Non-drug procedures, such as massage and other alternative therapies, can also be effective. Regardless of modality, patient-provider communication is essential for effective pain management. Support from family and loved ones is also instrumental, as supportive care entails numerous tasks central to health and treatment, such as transportation and monitoring of medication. Strong patient-provider and patient-family communication is critical to long-term survivorship.

For minority populations, the cultural constructs of health and illness beliefs may support or impede care. Complicating both the family and patient-provider relationships and conversations surrounding cancer symptom management are the survivor’s inability to express their pain experience in words [3], as well as traditional American Indian communication styles and preferences for non-verbal communication [4,5,6]. Both verbal and non-verbal communications are important in exchanges, and differences can lead not only to frustration, but also to misunderstanding. In addition, differing languages and cultures may create alternate perceptions of etiology, treatment needed for illness, and symptoms of cancer [7]. Currently, 573 American Indians and Alaska Natives are recognized by the Bureau of Indian Affairs (BIA), and many have a diversity of languages and may continue to speak their native language [8]. Although cancer has been long been documented in ancient remains [9], cancer and other health problems have remained in the population for over 500 years [10]. In addition, low health literacy and differing views of health and illness can lead to communication breakdowns between patients, caregivers, and healthcare providers [11].

Cancer rates, which were previously reported to be lower in American Indian populations, have been shown to be increasing during the past twenty years. Reported mortality rates are also higher than average. American Indians continue to experience the poorest survival from “all cancers combined” compared to all other racial groups [12]. Additionally, American Indian communities report a disproportionate rate of rarer cancers, as compared to the general population (i.e., elevated stomach and gallbladder cancer among Arizona and New Mexico Indians; elevated lung, colorectal, cervix, prostate, kidney, and stomach cancers for Alaska Natives; and elevated lung, prostate, and cervical cancer among Northern Plains Indians) [12]. Southwest American Indians face additional barriers of significant distance to care, due to many reservation sites being in remote, rural geographic locations.

Cancer screenings and medical treatment targeting American Indians have been increasingly offered over the past decades. Communicating the occurrence and stress of cancer-related pain and functionality to providers, caregivers, and family members is important so that timely and effective treatment can be realized. It is essential that focus group recording be transcribed verbatim to capture the participant’s voices, which informs the facilitator/researcher. Barkwell [13] discusses the interrelatedness of culture in the conceptualization of pain, noting that the perspectives of both the patient and the researcher hold equal validity and need to be legitimized (p. 243). Without these perspectives, understanding of the cancer survivor’s experiences are lost. For American Indian cancer survivors, pain and hardships that accompany the cancer diagnosis are seldom discussed. Such talk can be seen as a burden or a sign of weakness [3,6], thus many suffer in silence, which may have a negative impact on survivorship.

Culturally-bound influences, such as the cultural constructs of illness, cultural-prohibitive topics, and the desire for privacy [3] greatly influence the communication patterns of survivors. These factors contributed to the silencing of communications and reduced complaints regarding cancer diagnosis and cancer-related symptom management, including pain, fatigue, depression, and functionality. American Indians have experienced past traumas where tribes were decimated due to wars, illness, and disease, and their communities reside in poverty-stricken reservations and towns that report the poorest health, education, and economic status in the nation [10]. Persistent trauma within nuclear families and in communities burden the lives of many. Responses to these and more recent traumas may be encountered with stoicism and/or denial as a means of coping.

## 2. Results

Focus group findings were categorized into two major groupings, (a) communication with nuclear and extended family, and (b) communication with healthcare providers. Within these groupings of communication with families and providers, three themes emerged that described the cancer experience and cancer-related communication: (1) We don’t talk about it, (2) Respect for healthcare providers, and (3) Culturally prohibitive topics. Table 1 provides a description of coding and the identification of major and minor focus group themes.

### 2.1. We Don’t Talk about It

Cancer survivors explained that with regard to cancer, “we don’t talk about it:” a concise, yet comprehensive statement that encompasses a tangible barrier to communication about cancer, treatment, symptoms, or potential mortality. Survivors reflected on traditional American Indian communication, particularly preferences for silence. They reported that family members hid their cancer diagnoses and their cancer-related pain. They also did not talk about other cancer-related symptoms. This “stoic” behavior was often encountered among older or more self-identified “traditional” individuals residing in both rural and urban settings; however, younger individuals reported not discussing cancer or related symptoms as well under certain circumstances. For instance, participants reported that they would try to talk about cancer-related experiences and family members would remain silent. A family member shared, “he got diagnosed…like he was sick…we tried to ask him what he had…he wouldn’t say anything,” thus halting further communication. Another stated, “As for my sister, she was 55, she has been sick all this time but, she won’t even let us…nobody…talk about it.”

Upon probing within focus group sessions, several reasons for minimizing or restraining communication were presented. First and foremost, participants voiced that cancer survivors feel they have to be silent about their pain and not complain to family members so as not to produce stress or become a burden. One survivor said, “I just didn’t want them…to stress themselves over something I should be taking care of.” I was trying to keep that (diagnosis) from them, keep it away from them so they wouldn’t have to worry about me because they have their own life to take care of and I didn’t want anyone to get stressed over me.” Another stated she was in denial, “I don’t really want to talk about it. I haven’t accepted it.”

Some participants expressed the view that speaking about or acknowledging cancer would either bring it about or give cancer more power to cause more pain or hurt for everyone involved. One participant reported that she “kept it to herself, (although) I know she was in pain.” Emotional responses were also suppressed, because they may constitute disrespect by giving “acknowledgement” to the condition. For example, one participant stated, “We hide our tears: I know we hide our tears when my niece was going through (it) all.” A family member of a survivor commented, “Even families (who) visited after he was diagnosed—he stayed in bed. People did not talk about it.” Others also reported that they kept quiet—as “part of that respect, they keep to themselves. One of my uncles is like that…they don’t even go on that subject at all.” “I know my uncle admitted he knew there was something wrong. He was smoking. There was certain things that were happening. I think they know, but they don’t talk about it. I think they know but keep (it) to themselves.”

Although findings indicated that, for the most part, family members cooperated with survivors’ desires to keep communication closed, several participants in the study shared a desire for deeper communication. One participant stated, “They don’t talk about it. I wish we could talk to him.” A survivor’s nephew shared his attempt to talk to his uncle: “So, one day I saw he was at the feeding site…he lost a lot of weight. I asked him, ‘are you okay…are you sick?’ He knew what I meant, but he just kind of laughed it off, and said, ‘I am okay’… like that. But in the end when he was told that he had terminal cancer, he asked people to explain it to him, so I told him in Apache what the doctors said and what it meant and the things they told him. Later on, he just lowered his head and said, ‘adalezy,’ there is no hope. So he acknowledged it, but that was all he said about it” [14].

### 2.2. Respect for Healthcare Providers

In this study, a significant byproduct of deferential respect was incomplete communication with their providers, with many cancer survivors not disclosing the level of pain they were suffering. Several survivors stated they either did not want to complain, they had a fear of addiction, or the doctor did not ask them if they were in pain.

Focus group participants listed the following reasons for not telling their doctors that they are in pain:

(1) Fear of becoming addicted to pain medication: “I don’t want to rely on pain medications because of all the things people say about it too, that you can get hooked on it. I don’t want to do that.” Another said, “I have that fear of taking that pill.” When asked what are the fears you have, the survivor stated, “Because I might need it all the time, and that I might…” “Depend on it,” stated another survivor, completing her sentence.

(2) Doctor did not ask them if they were in pain: “He did not ask me.” Another survivor stated, “Yes, [pain] was always like eight. I was like man it’s eight. But today they didn’t ask me, and I was going to say I am right here smiling, laughing, it’s a three or a four.”

(3) They were not in pain at the moment of the doctor visit: “I told him today I feel fine, no joint pain, no headache, no tiredness.”

(4) They did not want to be a burden to their provider: “I guess I am not very comfortable to bring to anybody’s attention.” Another stated, “I didn’t want anyone to get stressed over me.” They also would neither request that their doctor clarify the meaning of medical terms nor their diagnosis. “I might ask them [a question]…but I don’t challenge them about it. I feel like that would be like a lack of trust.”

(5) They did not want to be a complainer, for example, “After my chemo treatments I go shopping with my daughter. I don’t want to give in,” and, “Well I kicked myself in the butt a couple of times, get over it, get over it.”

Results showed that participants took conversations with healthcare providers very literally—“I just go by their instructions,”—and essentially just wanted to make it through the visit, minimizing confrontation or opportunities for disagreement, for instance, “Well whatever they offer, whatever is going to do the job.” A survivor stated, “Whatever the doctors tell me. They know, they have their degrees, they know the sickness. It’s just my part in keeping my appointments, following through with what they tell me to do.”

### 2.3. Culturally Prohibitive Topic

Participants shared that they did not talk about cancer among family or friends as that was the same as talking about death (“that word cancer, cancer is death”), a culturally prohibited topic of discussion. When asked to explain why names of the deceased were not mentioned, the following excerpt explained the cultural prohibition against mentioning names of tribal members who have passed from cancer. One participant stated, “Our elderly tell us not to call them by the names (after death).” Another added, “They might come back…They are not of this world, they are in transition into the next world. It is pretty much you have respect for them…their footprints are no longer on this side. When it rains, their prints are erased. There is a term that is used…for someone that is passed on, if you use their name the elders will just get kind of appalled…yeah, Apache word ‘konachid’, meaning you dug them back up.”

Another participant added, “So I suppose that term itself wouldn’t exist in our language…it was the importance of respect.”

A combination of historical trauma, cultural constructs of illness beliefs, and communication patterns that limit or discourage open discussions formed the primary foundation for a legacy of silence. These illness beliefs also discouraged treatment-seeking behaviors that can lead to poorer prognosis. One participant responded, “Never did ask Dad what he feels after he was diagnosed. In the last three weeks he was going in and out of coma. In the last two weeks he was out of it. It was too sudden, I think they were just giving him medication.”

Seeing examples like this contributed to feelings among many in the study that cancer is meant to be and should just be accepted, as talking about it will not provide any relief or change in outcome.

## 3. Discussion

This study revealed culturally-bound communication restrictions between American Indian cancer survivors and their family. Cancer survivors’ response to a cancer diagnosis or cancer-related symptoms such a pain or loss of physical function is simply, “we don’t talk about it.” This response is a reaction to the cultural belief that discussing such experiences can bring additional pain, suffering, and hardship to the family or community. It also is a culturally prohibitive action, whereby talking about death (as “cancer is death”) may bring it to a reality. In addition, speaking the name of an individual who has passed is taboo—“they are not of this world, they are in transition into the next world” and one cannot call them back as it is “a matter of respect.” The result of these beliefs is a “suffering in silence” stance found among American Indian cancer survivors. This legacy of cancer-related silence experienced by parents and grandparents (often not exposed until after death) appears to be passed on to younger generations as well. For American Indians, silence is valued as a sign of respect, is used for emphasis, to allow autonomy of the speaker, and to convey unity or dissent, and yet to the untrained provider silence can be misconstrued as unengaged or not caring in Western culture. Particularly when considering stoicism in the face of a cancer diagnosis, a silent, culture-bound response to pain due to serious illnesses can be better understood. Silence as a coping mechanism may have resulted from the psychological consequence of trauma, poverty, and historical abuse endured by American Indian families and communities.

Cultural definitions of illness, perceptions of symptoms, treatment compliance, and health-seeking behaviors, as well as the delivery of health and prevention services [14] were important variables in this study. Culturally-bound illness beliefs and behaviors have a significant impact on mental health [15], thus awareness of the role of culture in communication, illness perceptions, and treatment beliefs, as well as the cultural value of self-reliance and the importance of respect and privacy among tribal groups were key topics of interest in this research. A common sentiment shared by the participants was that it was “too personal to talk about (cancer),” and “we don’t talk about it.” Talking about cancer or pain was taboo. Thus, survivors and family members alike are fearful that even saying the word “cancer” may bring forth the disease. “Because the fear…of the word itself …was something the community would not acknowledge.” Unfortunately, suffering in silence leaves many survivors less likely to seek preventative care or early treatment, as well as without much needed social and family support, contributing to reduced treatment compliance and poor access to healthcare.

For American Indians, hardships facing the individual, family, or community are often severe and experienced by several generations. Despite high cultural value placed on storytelling meant for teaching and sharing [16], dealing with cancer in the open appears discouraged; thus, willingly or unwillingly, cancer patients/survivors may suffer in silence. More research is therefore needed to determine the extent to which this influences patient/survivor outcomes. The reasons for suffering in silence, such as cancer-related pain experiences, can be due to culture-bound responses to hardship and illness beliefs, providing valuable insights for public health educators, tribal communities, health systems, and many others.

American Indian traditional health models balance the health of the mind, body, and spirit, each of which are related to the other—a concept that is often misunderstood by Western healthcare providers. Programs geared towards increasing cultural competence among healthcare providers are critical, not only to increase understanding of cultural differences between providers and patients, but also to build respect among providers for different communication and coping strategies. Cultural competence is an important part of the cancer survivor’s care. Respect is a key concept in the American Indian culture, not only in respect for the dead, but also for the living. Understanding the importance of respect held by the patient and the provider opens the door for shared decision-making. A key part of communication is respect, shared-decision-making, and cultural competency [17]. Survivors and family members in this study were split in their satisfaction with their care: some felt like they were on a conveyor belt and care was very impersonal, whereas others found their experiences to be fine, similar to a research study on American Indian/Alaska Native (AI/AN) veterans that found participants were “generally satisfied with the quality of care received” even though they “considered that the care they received was not fully culturally competent regarding AI/AN health beliefs and behaviors” [18].

Another issue of concern is low health literacy, a significant public health problem affecting vulnerable populations, including American Indians, particularly among the older generation [19]. The Patient Protection and Affordable Care Act defines health literacy as the degree to which an individual has the capacity to obtain, communicate, process, and understand basic health information and services to make appropriate health decisions [20]. Low health literacy can mystify a cancer diagnosis and complicate treatment instructions and medication regiments, often overwhelming cancer patients and their families and caregivers. Patients with less understanding of their disease are less able to analyze the barrage of information, make informed decisions, or effectively manage their symptoms. According to the Indian Health Service’s Health Literacy Workgroup, “improving communication between patients and health care providers is one of the best ways to combat low health literacy” [19] (p. 6). However, family members and caregivers may also have low health literacy and improving family and community health literacy is an important aspect of healthcare. Meeker [21] reports that caregivers with higher pain management knowledge had significantly fewer barriers to cancer pain management, supporting the importance of increasing caregivers’ knowledge of management of cancer pain. Family members and caregivers are integral to cancer survivors’ health and can serve as important advocates for pain mitigation and control during healthcare visits.

## 4. Materials and Methods

Thirteen focus groups, composed of nine to eleven participants in each group, were held over a two-year period at three urban and two rural reservation sites in the state of Arizona. Adult American Indian cancer survivors (often accompanied by their caregivers) were recruited by informational flyers (as recommended by local tribal personnel) placed in American Indian healthcare clinics and in community-wide facilities such as day care and community meeting rooms. The sessions were designed to inform a larger randomized trial that tested a cancer symptom management intervention [22]. Better understanding of American Indian cancer survivors’ cancer experience was sought to inform the larger study’s baseline questions and measures on cancer experiences, pain management knowledge, network of support, and cancer care needs. For this study, we utilized self-reporting as a measure for assessment of pain for survivor appraisal/observational assessment. Research [23,24] indicates that communication is difficult due to a variety of personal (i.e., inadequately assessed) and interpersonal factors, which can lead to inadequate information and poor pain management. As such, self-reporting of the survivors’ pain was targeted and carefully audio-recorded and transcribed verbatim.

Study inclusion criteria included: (a) self-identified American Indian, (b) cancer survivor following a physician’s diagnosis, (c) age 18+ years, and (d) southwest resident. The two-hour oral focus group sessions, comprising approximately 10 participants each, were audio-recorded and transcribed verbatim. Language transcribers from the Apache and Navajo Nation were secured to assist with the transcription of the audiotapes. Incentives in the form of gift cards ($35) covered travel and expenses. Two UCLA faculty researchers trained in focus group design and implementation facilitated the focus groups. Paper questionnaires were not administered and identifiers (such as age, gender, cancer diagnosis) that would breach confidentiality were not collected for this portion of the study. Focus group members were asked a series of questions regarding their experiences, symptom management, and communication regarding their cancer diagnosis: These questions included:Their experiences with cancer. (“What does your sickness do to you, your body, your life? What are the chief problems that your sickness has caused you?”).Family interactions/communication regarding cancer diagnosis and cancer-related symptom management. (“What does cancer do to your family? How does it affect your daily living?”).Impact of specific symptoms (such as pain, fatigue, loss of function, depression).Treatment, treatment preferences and needs (“How is your sickness treated? How do you think it should be treated? What other things do you do to treat it? What kinds of assistance do you need?”).Cancer-related stigmas and cultural barriers (“Are there any taboos or stigmas associated with your illness?”).

Institutional Review Board (IRB) approvals were obtained from the University of California, Los Angeles (UCLA) and the Phoenix Area Indian Health Service (via the Phoenix Indian Medical Center IRB). Approvals were also obtained from each tribe and urban Indian center who participated in the study. Participants consented prior to their enrollment in the focus groups. A copy of the consent form was provided to each participant. They were told their participation was voluntary, healthcare services were not contingent upon participation, and that they did not have to answer any question(s) that they did not want to or those that they found uncomfortable. Further, participants were assured that their responses were confidential, as their name and other identifiers would not be recorded. Participant code assignment helped to ensure confidentiality.

Focus group sessions were held in tribal community meeting rooms or in private rooms in Indian health clinic facilities. Arrangements were made for bottled water, as well as coloring pencils and papers for children to use in the waiting rooms if needed. Caregivers were also allowed to accompany the participant if needed and/or preferred.

Following each focus group, the tape records and facilitator notes were gathered and returned to the UCLA research office for analysis. Once transcribed verbatim, the tapes were shredded to maintain confidentiality. The research team used constant comparative methods in the analysis of the dialogues. Transcripts were read in their entirety and then excerpt-by-excerpt coding was conducted to capture the meaning expressed in each excerpt. A qualitative data analysis software (ATLAS.ti, version 7, Berlin, Germany) [25] was used to categorize data and to manage the developing themes. Codes were grouped by importance, related concepts, and frequency as major themes emerged from the data. Another investigator and two American Indian translators (one Navajo and one Apache) independently reviewed tapes, as well as the categorized codes and key themes, for corrections due to language or cultural constructs. Identified categories, grouped for comparison and the final assessment, used constructivist Grounded Theory methods [26] to analyze the data.

Both rural reservation sites and urban Indian clinics were sampled and each group represented a sampling of current or former cancer patients and their caregivers residing in each location. As identifiers were not collected it was not possible to strategize the groups by age, gender, or cancer type. Out of the 13 groups, 29.4% were male participants, 70.6% female, 30.2% were patients, and 69.8% were family members/caregivers. Overall, for each site there was a majority of female participants. Self-reporting methods were utilized for patient pain assessment and self-reported observational report for caregivers.

Although some selection bias may be present due to the small sample size, recruitment methods were approved by each tribal site and health clinic site and were considered to be the least invasive and most effective for this phase of the study. This manuscript’s contents are solely the responsibility of the authors and do not necessarily represent the official views of the NIH/NCI or the Indian Health Service.

## 5. Conclusions

Pain in cancer survivors is a common yet rarely talked about symptom amongst American Indians. A systematic review by Jimenez [27] reports Americans Indians have a high incidence of pain coupled with poor patient-provider communication, resulting in a significant impact on pain assessment and management. Our study identifies three major premises for the silence observed in American Indians: (1) we don’t talk about it, (2) respect for healthcare providers, and (3) culturally prohibitive topics of death and pain experiences. Closed communication about cancer experiences leaves cancer survivors with diminished social support and places them at risk for reduced treatment compliance and compromised access to healthcare. These findings have implications for educational interventions and quality of life improvement for American Indian and other underrepresented communities. We recommend culturally appropriate interventions that take into consideration the cultural constructs of cancer and cancer pain, the difficulties associated with patient-provider communications, and the many big and small barriers in cancer care, which include access to care, transportation, and improved educational interventions. Hearing the voices of American Indian cancer survivors provides rich information data for program development and implementation. The cultural constructs of cancer experiences and cancer pain management need to be legitimized through programs designed especially for American Indians.

## Figures and Tables

**Table 1 cancers-12-01932-t001:** Description of coding, identification of major and minor focus group themes.

Coding Category	Major	Minor
We don’t talk about it	Stoic behavior Hides cancer symptoms Expected to bear things like pain Denial	Don’t bring burden home Speaking about it gives it more power or brings on cancer Talking about cancer is like talking about death Disrespectful to talk about cancer
Respect for healthcare providers	Silence is sign of respect in patient-provider communication Incomplete communication—not disclosing level of pain Fear of addiction to pain medications Minimizing confrontation	Do not shame provider by correcting errors or asking too many questions Not wanting to be a burden to providers Not always in pain Does not want to be a complainer
Culturally prohibitive topic	Cannot talk about death Fear of the unknown Cancer diagnosis is a death sentence	The deceased may come back Passively accepts diagnosis No survival from cancer

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
