# Peer review of "We Don’t Talk about It: Cancer Pain and American Indian Survivors"

_cancers, 2020, doi:10.3390/cancers12071932_

Round 1

Reviewer 1 Report

The authors have done a very nice report of focus groups with Native Americans regarding cancer pain and survivorship. This manuscript confirms the findings from a systematic review (Jiminez J Pain, 2012) that Native Americans have a high incidence of pain and poor patient-provider communication which impairs both the assessment of pain and the delivery of adequate pain management.

This manuscript would benefit from the inclusion of a few other references that have incorporated more cultural competency. Blackwell (J Pain & Sym Manage 2005) looks at the voices of the Ojibway people and provides additional context for the differences in perspectives from those running the focus groups and those providing responses through the focus groups. In addition, Haozous (Oncol Nursing Forum 2016) provided results from focus groups of Native American cancer survivors that incorporated Native American methods and the investigators provided a conceptual framework that grew out of the Native American culture and provided a means of hope to the survivors.

This manuscript is a well written description of focus groups to better understand the Native American perspective, but the discussion does not provide any "hope" to the reader to understand how best to address the issues that were raised. 

Reviewer 2 Report

The authors presented an interesting qualitative research about cancer pain issues among a vulnerable populations. The three main issues are important and discussions are adequate. However, there are several major issues the authors should address:

1) there were thirteen focus groups (N=132). Could the authors describe the distributions of age, gender, and cancer types  and time since diagnosis of the participants? Was each group targeting certain age/gender/cancer type, or how mixed each group was? 

In addition, young and old people, male and female may have different issues in cancer pain. Long term cancer survivors and recent cancer survivors may have different issues. The authors should present and discuss these issues.

2) the authors identified three themes and several major and minor issues. What basis and methods did authors use to classify them? It is desirable to have some quantitative information about each issue. For example, out of 13 groups, how many groups discussed or emphasized each major issue or each minor issue?

3) What are the theoretical framework underlying the pain communication issue?

4) Compared with other racial/ethnical groups, what are the unique issues of cancer pain in the Southwest American Indians?  There are several other large tribes in the US. Are these issues unique to Southwest American Indians or common in all American Indians?

5)  Another factor is the life style. Given that this is a qualitative study with small sample size, it is possible that the selected participants are not well representing the population. What is the socio-economic status of the participants? The recruitment is through flyers. Will there be some selection bias with this recruitment?

Overall, I think this is an interesting paper, but much details should be provided.

Reviewer 3 Report

Please see the attachment (comments in PDF file) as well as additional comments below.

Several citations either appeared out of order and/or did not support statements by the co-authors. 

Make sure if the two reservations where the study was conducted included approvals at the local level (resolutions) and not rely on Phoenix Area Indian Health Service. The reviewer wonders why the ITCA was not utilized for approval. If Navajo Nation was included in the study, make sure Navajo Nation Human Research and Review Board has approved the study (approval does not include urban Navajos). 

Were the translators (Apache and Navajo) given information about confidentiality? Not to share information to others? Did they completed a CITI training?

The reviewer wonders why Dr. Linda Burhansstipanov's work on cancer survivor was not cited. She is a well known cancer researcher, especially cancer survivors among AIs. 

Author Response

For comments left in PDF file, I cannot thank you enough for bringing to my attention the errors in references and other manuscript discussions that needed clarification or revisions. I have revised the manuscript and worked to respond to your second review and I do hope I have covered all of the concerns.

Please also see the attachment.
